# Variations in the Visual Probe Paradigms for Attention Bias Modification for Substance Use Disorders

**DOI:** 10.3390/ijerph16183389

**Published:** 2019-09-12

**Authors:** Melvyn Zhang, Daniel S. S. Fung, Helen Smith

**Affiliations:** 1National Addiction Management Service, Institute of Mental Health, Singapore 539747, Singapore; 2Family Medicine & Primary Care, Lee Kong Chian School of Medicine, Nanyang Technological University Singapore, Singapore 308232, Singapore; h.e.smith@ntu.edu.sg; 3Institute of Mental Health, Department of Developmental Psychiatry, Singapore 539747, Singapore; daniel_fung@imh.com.sg

**Keywords:** Visual Probe Task, attention bias, cognitive bias, psychiatry

## Abstract

Advances in experimental psychology have provided evidence for the presence of attentional and approach biases in individuals with substance use disorders. Traditionally, reaction time tasks, such as the Stroop or the Visual Probe Task, are commonly used in the assessment of attention biases. The Visual Probe Task has been criticized for its poor reliability, and other research has highlighted that variations remain in the paradigms adopted. However, a gap remains in the published literature, as there have not been any prior studies that have reviewed stimulus timings for different substance use disorders. Such a review is pertinent, as the nature of the task might affect its effectiveness. The aim of this paper was in comparing the different methods used in the Visual Probe Task, by focusing on tasks that have been used for the most highly prevalent substance disorders—that of opiate use, cannabis use and stimulant use disorders. A total of eight published articles were identified for opioid use disorders, three for cannabis use disorders and four for stimulant use disorders. As evident from the synthesis, there is great variability in the paradigm adopted, with most articles including only information about the nature of the stimulus, the number of trials, the timings for the fixation cross and the timings for the stimulus set. Future research examining attentional biases among individuals with substance use disorders should take into consideration the paradigms that are commonly used and evaluate the optimal stimulus and stimulus-onset asynchrony timings.

## 1. Overview of Attention Bias Assessment and Modification

Advances in experimental psychology have provided evidence for the presence of attentional and approach biases in individuals with substance use disorders. Attentional biases result in individuals having a preferential allocation of their attentional processes to substance-related stimuli [1], while approach biases result in individuals having automatic action tendencies in reaching out for substance-related cues [2]. Various theories have provided explanations for the presence of these biases, including that of the incentive-sensitization theory, the classical conditioning theory and that of the dual-process theory [3]. The dual-process theory is most commonly used in the justification of the presence of attentional biases. It postulates that the repeated use of a substance would result in increased automatic processing and increased automatic tendencies to approach substance-specific cues, with the inhibition of normal cognitive control processes [3]. The discovery and the understanding of these unconscious, automatic biases are of importance clinically, as they help to account for the lapses and relapses among individuals with substance use disorders [4]. Recent neuroimaging studies have highlighted that attentional biases are associated with increased activation in several neuroanatomical regions, including that of the anterior cingulate cortex, dorsolateral prefrontal cortex, insula, nucleus accumbens and amygdala [5,6].

Traditionally, reaction time tasks, such as the Stroop or the Visual Probe Task, are commonly used in the assessment of attention biases [1]. In the Stroop task, individuals are required to name the colours of both the neutral and the drug-related words. In the Visual Probe Task, participants are required to respond readily to a probe that would replace either the neutral or the drug image. Attentional biases are deemed to be present if individuals respond more readily to a probe or words that replace substance-related images, as compared to the neutral word or images [1]. Ataya et al. (2012) [7] previously reported that both these tasks (the Stroop and Visual Probe Task) are associated with poor internal reliability. Field et al. (2012) [8] have postulated that one of the factors that could account for the poor reliability of the Visual Probe Task pertains to the type of stimulus used. Thus, the authors proposed the use of a personalised stimulus and images that participants could readily identify with. In turn, this could then result in a more demonstrable change in biases [8]. In a review by Lopes et al. (2015) [9], they reported that the Visual Probe Task was effective—in 88% of the studies involving individuals with substance use disorders, there was successful retraining of attentional biases. Jones et al. (2018) [10], in their recent study, have explored methods to improve the internal reliability of the Visual Probe Task. The authors examined the nature of the stimulus included, adopting the previous suggestion of having a personalised stimulus and reported that the inclusion of a personalised stimulus did help to improve the internal consistency of the Visual Probe Task.

Nevertheless, the authors still report the Visual Probe Task to be unreliable and that reliabilities were acceptable if the stimulus cues were presented for short intervals. Jones et al. (2018) [10], in their review, highlighted that there is great variation in the timings of the stimulus intervals used in published studies involving addiction. However, a gap remains in the published literature, as there have not been any prior studies that have reviewed the stimulus timings for the different substance use disorders. Such a review is pertinent, as the nature of the task might affect its effectiveness. Given this, our aim was to compare the different task paradigms and methods for the Visual Probe Tasks used for the most highly prevalent substance disorders—that of opiate use, cannabis use and stimulant use disorders. 

## 2. Visual Probe Trask Paradigms in Published Studies

Two recent reviews have synthesised the evidence for attentional biases among substance users. Maclean et al. (2018) [11] identified 21 studies that have previously examined attentional biases in opioid using individuals. Zhang et al. (2018) [4] identified 11 articles involving participants with opioid use disorder, 16 articles with participants with stimulant use disorders and nine articles involving participants with cannabis use disorders. In order to fulfil our aim, we will describe the Visual Probe Task paradigms (the methods of the Visual Probe Tasks) that have been used in each of the published studies. 

Table 1 provides an overview of the characteristics of the Visual Probe Task that were utilized in previous studies involving individuals with opioid use disorders. From both Maclean et al. (2018)’s [11] and Zhang et al. (2018)’s [4] review, we managed to identify a total of eight articles that specified the use of the Visual Probe Task for attention bias assessment or modification. We were unable to access the full text of one of the journals as it was published in a Chinese Journal. In the identified articles, there was great variability in the number of stimulus included, ranging from 12 to 44 picture pairs. Some studies included as few as 64 trials [12,13], while others included as many as 512 trials [14]. Across the studies, there was great variation in the Visual Probe Task. Most of the studies presented the fixation cross for 500 ms, except Frankland et al. (2016) [15] and Zhao et al. (2017) [16], who presented the fixation cross for 1000 ms. Several studies have presented the stimulus and neutral image set for both a short and long duration [12,13,14,15,17,18]. The short stimulus timing was commonly that of 200 ms, and the long stimulus timing was that of 2000 ms, though, in Frankland et al. (2016)’s [15] study, they presented the images for 500 and 1500 ms as well.

While all the studies explicitly stated that they were based on the Visual Probe Task, there were variations in the nature of the task. Some studies [12,13,18] have included an interstimulus interval, before the presentation of the probe. Also, in some studies, the probe remained on the screen until the participant made a response [14,16,17], while in other studies, the probe only appeared for 100 ms, before disappearing [12,13,18]. Some studies also included an inter-trial interval, but there was variation in the timing of this interval (from 250 to 2000 ms). Some of the studies [15,16,19] have included practice trials. Figure 1 provides a graphical representation of the details of the Visual Probe Task that were reported in each of the identified studies for opioid use disorder. 

Table 2 provides an overview of the characteristics of the Visual Probe Task that was utilized in previous studies involving individuals with cannabis use disorders. A total of three articles from Zhang et al. (2018)’s [4] prior review were included, as they have reported the use of the Visual Probe Task. For the stimulus, Field et al. (2004) [21] included words instead of pictorial stimuli. In terms of the Visual Probe Task, two studies [21,22] presented the fixation cross for 500 ms, whereas Field et al. (2006) [23] presented it for 1000 ms. There was again variation in the timings for the stimuli cues, with two studies [21,22] presenting the stimulus cue for 500 ms, whereas that of Field et al. (2006) [23] presented it for 2000 ms. In terms of probe presentation, two studies presented the probe [21,22] until a response was made. Vujanovic et al. (2016) [22], presented the probe for 125 or 250 ms. Across all the studies, they have included an inter-trial interval, which ranged from 1000 to 1500 ms. In terms of the number of trials, it ranged between 72 and 96. All the studies included practice trials for participants. Figure 2 provides a graphical representation of the details of the Visual Probe Task that were reported in each of the identified studies for cannabis use disorder.

Table 3 provides an overview of the characteristics of the Visual Probe Task that were utilized in the previous studies involving individuals with stimulant use disorders—that of cocaine use disorders. A total of four articles from Zhang et al. (2018)’s [4] prior review was included. While all the studies have their basis in the Visual Probe Task, there was variability in the paradigms. Some studies included 10 sets of images [24], while others [25,26] included up to 20 sets of images. There was variation in the number of trials individuals had to undertake, ranging from 80 to 240 trials. Two studies [25,26] reported the inclusion of practice trials. Three out of the four studies reported that they presented a fixation cross for 500 ms. In terms of stimulus timings, they were presented for 500 ms in three studies [24,25,26] and for a short (200 ms) and long (500 ms) interval in Mayer et al. (2016)’s study [27]. In all of the studies, the probe appeared up until a response was made. Only two of the four identified studies allowed for an inter-trial interval [24,27]. There was variation in the inter-trial interval, as it ranged from 500 to 1500 ms. Figure 3 provides a graphical representation of the details of the Visual Probe Task that were reported in each of the identified studies for stimulant use disorder.

## 3. Implications for Future Research

It is apparent that there is great variability in the paradigm of the Visual Probe Task. In addition, there is also a varied amount of information shared about the nature of the paradigm. Most of the articles included information about the nature of the stimulus, the number of trials, the timings for the fixation cross and the timings for the stimulus set. However, information is missing in some studies, with regards to the inter-stimulus interval, the time that the probe appears for, the inter-trial interview and the time allocated for the individual to response. The absence of this information limits the reproducibility of the Visual Probe Task by others. For future research, it is essential that the intervention is described in full, carefully specifying the full methodology of the Visual Probe Task used, in order to allow for the replication of studies.

While there were clear variations in the paradigms, there were some common elements across all the studies. For studies involving participants with opioid use disorders, most of the studies presented the fixation cross for 500 ms and presented the set of stimulus images for both a short and long stimulus duration. A shorter stimulus duration would allow for the evaluation of the initial, automatic detection attentional processes, while a longer stimulus duration would allow for the evaluation of the engagement stages of attention [9]. In contrast, for studies involving participants with cannabis use or stimulant use disorders, the stimulus pair was most commonly presented for 500 ms. Most of the studies utilizing these timings have provided positive findings for attentional biases, except for Charles et al. (2015) [14] and Mayer et al. (2016) [27].

This evidence synthesis has direct implications for future research. We propose that future studies assessing and modifying attentional biases among individuals with opioid use disorders should consider the use of both a short and long stimulus timing, whereas studies evaluating attentional processes among people using cannabis or with stimulant disorders could use a single stimulus interval. To date, there is only a single study (Mayer et al., 2016) [27] that has examined a varying timing stimulus for individuals with stimulant use disorder. Future research should also examine whether the presence of a varying stimulus timing interval will enhance the detection and modification of attentional biases among individuals with cannabis and stimulant use disorders.

From the studies that we have included, there were a limited number of studies that have reported the stimulus-onset asynchrony timings. Lopes et al. (2015) [9], in their prior review exploring the Visual Probe Task for various disorders, have reported that there was variation in the timings for the different psychiatric disorders. For substance use disorders, it ranged from 50 to 500 ms; for depressive disorders, it ranged from 500 to 2000 ms; and for anxiety disorders, it ranged from 200 to 1500 ms. Lopes et al. (2015) [9] have previously highlighted that a relatively longer stimulus duration is advantageous as it allows for participants to fully process the nature of the stimuli. However, as evident from this evidence synthesis, there are few studies that report on this timing, and this is indeed an area that future research should evaluate to determine the optimal interval for the different substance use disorders.

Our article is perhaps the first article that has reviewed the task paradigms that have been adopted in previously published studies, involving individuals with substance use disorders. We managed to systematically extract information, primarily from the methods section of each manuscript, to ascertain the details of the visual probe paradigms that were utilized. However, there were limitations in our current study. We were unable to access the full text of one of the published articles, as it was published in a Chinese Journal. We have also attempted to contact each of the authors for further details about the Visual Probe Task they have previously used but, to date, we have only managed to receive replies from a single author, who stated that all the details have already been cited in the methods section of the published manuscript.

## 4. Conclusions

Our article has reviewed all the visual probe paradigms that have been applied previously for addiction and substance research. While there are variations in the underlying paradigms, there are some commonalities as well. Future research examining attentional biases among individuals with substance use disorders should take into consideration the paradigms that are commonly used and evaluate the optimal stimulus and stimulus-onset asynchrony timings.

## Figures and Tables

**Figure 1 ijerph-16-03389-f001:**
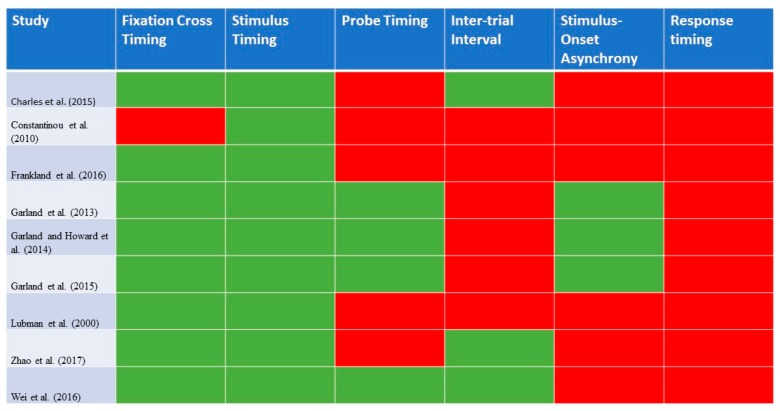
A graphical representation of the details of the Visual Probe Task that were reported in each of the identified studies for opioid use disorder (n = 8). Green highlights: reported in study; red highlights: not reported in study.

**Figure 2 ijerph-16-03389-f002:**
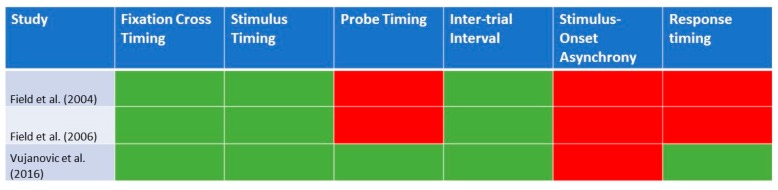
A graphical representation of the details of the Visual Probe Task that were reported in each of the identified studies for cannabis use disorder (n = 3). Green highlights: reported in study; red highlights: not reported in study.

**Figure 3 ijerph-16-03389-f003:**
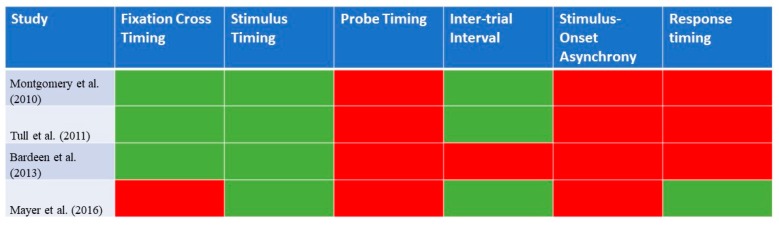
A graphical representation of the details of the Visual Probe Task that were reported in each of the identified studies for stimulant use disorder (n = 4). Green highlights: reported in study; red highlights: not reported in study.

**Table 1 ijerph-16-03389-t001:** Characteristics of the Visual Probe Task used in previous studies involving individuals with opioid dependence (*n* = 8).

Study	Participants	Intervention Details	Nature of Stimulus Included	Details of Assessment Task	Outcomes
Charles et al. (2015) [14]	In total, 23 opiate users and 21 healthy controls	Dot Probe Task for attention retraining in the lab. Participants maintained on Buprenorphine and Methadone Maintenance Therapy (MMT)	In total, 44 picture pairs In total, 40 pairs matched for visual complexity and composition, contained opiate-related items (spoons, needles, lighters, heroin-like substance) The other four pairs were neutral images only	Fixation Cross Timing: 500 ms Stimulus Pair Timing: Short (200 ms) or Long (500 ms) Probe: Arrow pointing upwards or downwards, remained on screen until participant responded Probe Timing: Not mentioned Inter-trial interval: 250–500 ms Stimulus Onset Asynchrony (SOA): Not mentioned Response Timing: Not mentioned Total number of critical trials: 64 (assessment), 512 (Intervention) Total number of neutral trials: 16 Total number of practice trials: Not mentioned	No baseline difference in attention bias between patients and controls No significant effect of Attention Bias Modification (ABM) on Attention Bias (AB) or substance cravings
Constantinou et al. (2010) [17]	In total, 16 opiate users in treatment, 16 ex-users, and 16 healthy controls	Dot Probe Task for attention retraining in the lab. Participants maintained on methadone maintenance therapy	In total, 40 picture pairs Of which, 20 pairs matched for visual complexity and composition, and contained one opiate related picture and one non opiate related picture The other 20 pairs were neutral pairs	Fixation Cross Timing: Not mentioned Stimulus Pair Timing: Short (200 ms) or Long (2000 ms) Probe: Probe appeared on the side of the screen where one of the pictures have been previously Probe Timing: Not mentioned Inter-trial interval: Not mentioned SOA: Not mentioned Response Timing: Not mentioned Total number of critical trials: 160 Total number of neutral trials: Not mentioned Total number of practice trials: Not mentioned	Greater attentional biases towards drug-related stimuli for current users, as compared to ex-users. Ex-users showed a bias away from drug-related stimuli in the stress condition and this correlated positively with their length of abstinence
Frankland et al. (2016) [15]	In total, 19 were opioid dependent and 20 healthy controls	Dot Probe Task for attention retraining in the lab. Participants maintained on partial MMT	In total, 14 drug-related images (images of drug paraphernalia and an unidentified addict appearing to cook up and inject heroin), Another 14 pairs matched for control pictures (items from a children’s building game, and a person building a model railway)	Fixation Cross Timing: 1000 ms Stimulus Pair Timing: 200, 500 and 1000 ms Probe: Probe replaced the images. Participants to press one of the buttons as quickly as quickly as possible, without making mistakes, to indicate whether the probe appeared on the left- or right-hand side. Probe Timing: Not mentioned Inter-trial interval: Not mentioned SOA: Not mentioned Response Timing: Not mentioned Total number of critical trials: 164 Total number of neutral trials: 84 Total number of practice trials: 10	Opioid dependent group had a significant attentional bias for opioid related information presented at 200 ms and 500 ms. No attentional biases at 150 0ms.
Garland et al. (2013) [13]	In total, 32 were opioid dependent and 33 non opioid-dependent	Dot Probe Task for attentional retraining in the lab Participants maintained on partial buprenorphine and MMT	In total, 12 opioid images, including photos of pill (Oxycontin, Vicodin), pill bottles, crushed and powdered opioids for insufflation, and a syringe next to a vial of injectable morphine Neutral images included 12 photos from the International Affective Picture System, depicting household items	Fixation Cross Timing: 500 ms Stimulus Pair Timing: 200 or 2000 ms Probe: Target probe will replace the images after 50 ms inter-stimulus interval Probe Timing: 100 ms Inter-trial interval: Not mentioned SOA: 50 ms Response Timing: Not mentioned Total number of critical trials: 64 Total number of neutral trials: 12 Total number of practice trials: Not mentioned	Opioid-dependent individuals had significant attention bias towards opioids cues presented for 200 ms but not for cues presented for 2000 ms
Garland and Howard et al. (2014) [12]	In total, 28 with a high risk for misuse and 19 with a low risk for misuse	Dot Probe Task for attentional retraining in the lab	In total, 12 opioid images, including photos of pill (Oxycontin, Vicodin), pill bottles, crushed and powdered opioids for insufflation Neutral images included 12 photos from the International Affective Picture System (IAPS), depicting household items	Fixation Cross Timing: 500 ms Stimulus Pair Timing: 200 or 2000 ms Probe: Target probe will replace the images after 50ms inter-stimulus interval Probe Timing: 100 ms Inter-trial interval: Not mentioned SOA: 50 ms Response Timing: Not mentioned Total number of critical trials: 64 Total number of neutral trials: 12 Total number of practice trials: Not mentioned	Biased initial attentional orienting to prescription opioid cues
Garland et al. (2015) [18]	In total, 72 opioid misusers and 26 opioids non misusers	Dot Probe Task for attentional retraining in the lab	Pair of photos containing one emotionally salient image and one neutral image Three blocks of cues (opioid-related, pain-related, and pleasure-related cues) Opioid-related cues included images of pills and bottles Pain-related cues included images of severe injuries, painful medical procedures, and human faces grimacing in pain Natural reward cues included images of romantic couples and food In total, 36 neutral images selected from International Affective Picture System (IAPS)	Fixation Cross Timing: 500 ms Stimulus Pair Timing: 200 or 2000 ms Half of the trials presented at 200ms, half 2000 ms. Probe: Target probe will replace the images after 50 ms inter-stimulus interval Probe Timing: 100 ms Inter-trial interval: Not mentioned SOA: 50 ms Response Timing: Not mentioned Total number of critical trials: 64 Total number of neutral trials: 12 Total number of practice trials: Not mentioned	Opioid misusers exhibit attentional deficits during reward processing
Lubman et al. (2000) [19]	In total, 16 methadone-maintained addicts and 16 age-matched controls	Pictorial Probe Detection task	Drug-related photographs (drug paraphernalia, for example, needles, spoons and heroin wraps) Scenes of unidentified addict cooking up and injecting heroin. Control photographs included items from children building a game and scenes showing an unidentified person building a model railway using various components from the set	Fixation Cross Timing: 1000 ms Stimulus Pair Timing: 500 ms Probe: After each pair of images, a dot probe will appear in the position of one of the pictures Participants are to press as quickly as possible, one of the two response buttons. Probe Timing: Not mentioned Inter-trial interval: Not mentioned SOA: Not mentioned Response Timing: Not mentioned Total number of critical trials: 160 Total number of neutral trials: Not mentioned Total number of practice trials: 12	Attentional biases in opiate addicts to probe that replaced the drug pictures, rather than neutral images
Wei et al. (2016) [20]	In total, 22 heroin addicts and 22 healthy controls	Visual Probe Task	In total, 10 heroin images The other 10 were natural scenery pictures	Fixation Cross Timing: 1000 ms Stimulus Pair Timing: 500 ms Probe: Appear in the location where one of the images disappear Probe Timing: 200 ms Inter-trial interval: 1350 ms SOA: Not mentioned Response Timing: Not mentioned Total number of critical trials: 80 and 240 Total number of neutral trials: Not mentioned Total number of practice trials: 20	Heroin addicts have more rapid response when the dot located on the heroin-related picture as compared to the neutral picture
Zhao et al. (2017) [16]	In total, 30 methadone-maintained outpatients and 38 healthy controls	Visual Probe Task	Drug neutral picture pairs and neutral–neutral picture pairs. In total, 20 pictures of substance-related scenes and 20 similar pictures matched with same layout but lacking substance-related cues Another 20 pairs of pictures matched with similar neutral scenes of daily life	Fixation Cross Timing: 1000 ms Stimulus Pair Timing: 2000 ms Probe: Probe then displayed in one of the positions of the pictures Probe remained until the participant responded by clicking the left or right button of mouse Probe Timing: Not mentioned Inter-trial interval: 200 ms SOA: Not mentioned Response Timing: Not mentioned Total number of critical trials: 120 Total number of neutral trials: Not mentioned Total number of practice trials: 16	Heroin group reacted faster to probes associated with substance-related pictures than neutral pictures More initial fixations Maintained longer initial fixation durations towards substance-related pictures

**Table 2 ijerph-16-03389-t002:** Characteristics of the Visual Probe Task used in previous studies involving individuals with cannabis dependence.

Study	Participants	Intervention Details	Nature of Stimulus Included	Details of Assessment Task	Outcomes
Field et al. (2004) [21]	In total, 17 regular cannabis users and 16 non-users	Visual Probe Task	Cannabis-related words, environment-related words, pleasant words, and unpleasant words	Fixation Cross Timing: 500 ms Stimulus Pair Timing: 500 ms Probe: Immediately after the offset of the words, a small dot probe was presented in the position of one of the words, until the participant gave a manual response. Probe Timing: Not mentioned Inter-trial interval: 1000 ms Stimulus Onset Asynchrony (SOA): Not mentioned Response Timing: Not mentioned Total number of critical trials: 64 Total number of neutral trials: 32 Total number of practice trials: 12	High levels of craving associated with significant attention bias for cannabis-related words
Field et al. (2006) [23]	In total, 23 regular cannabis users and 23 non-user controls	Visual Probe Task with concurrent eye movement monitoring	In total, 18 cannabis-related photographs Depicting a scene relating to cannabis use Control photograph (did not contain cannabis related content)	Fixation Cross Timing: 1000 ms Stimulus Pair Timing: 2000 ms Probe: After picture offset, small visual probe (an arrow which is pointed up or down) was presented on the left or right of the screen Respond to the probe as quickly as possible Probe Timing: Not mentioned Inter-trial interval: 500 ms SOA: Not mentioned Response Timing: Not mentioned Total number of critical trials: 72 Total number of neutral trials: Not mentioned Total number of practice trials: 10	Regular users had biases to maintain gaze on cannabis cues and faster approach responses to cannabis cues
Vujanovic et al. (2016) [22]	In total, 12 adults with cannabis use disorder and 13 controls	Visual Probe Task	In total, six cannabis pictures—cannabis-related stimuli— and six neutral pictures matched in size, colour and context	Fixation Cross Timing: 500 ms Stimulus Pair Timing: 500 ms Probe: Probe stimulus removed, picture (cue) remained on screen 1500 ms Probe Timing: 125 or 250 ms Inter-trial interval: 1500 ms SOA: Not mentioned Response Timing: 1500 ms Total number of critical trials: 96 Total number of neutral trials: Not mentioned Total number of practice trials: 12	Cannabis use group showed greater attentional biases to cannabis cues at the 125 ms probe time

**Table 3 ijerph-16-03389-t003:** Characteristics of the Visual Probe Task used in previous studies involving individuals with stimulant dependence.

Study	Participants	Intervention Details	Nature of Stimulus Included	Details of Assessment Task	Outcomes
Montgomery et al. (2010) [24]	In total, 32 regular cocaine users and 40 nonusers	Visual Probe Task and Modified Stroop Task	In total, 10 pairs of images, with one cocaine-related image depicting cocaine, cocaine paraphernalia, or close up of an individual using cocaine Matched with a neutral image, perceptually similar to cocaine image as possible, but did not have cocaine content	Fixation Cross Timing: 500 ms Stimulus Pair Timing: 500 ms Probe: Visual probe (an arrow pointing up or down) presented in the location previously occupied by one of the pictures Required to rapidly identify the orientation of the arrow probe by pressing the appropriate arrow on the keyboard Probe Timing: Not mentioned Inter-trial interval: 500 ms Stimulus Onset Asynchrony (SOA): Not mentioned Response Timing: Not mentioned Total number of critical trials: 80 Total number of neutral trials: Not mentioned Total number of practice trials: 10	Cocaine participants who consumed alcohol had increased attentional biases for cocaine pictures
Tull et al. (2011) [26]	In total, 30 cocaine-dependent patients with Post-traumatic Stress Disorder (PTSD) and 30 cocaine-dependent patients without PTSD	Visual Probe Task	In total, 20 cocaine-related images (crack rocks, powder cocaine, crack pipes, etc.) and 40 images of furniture	Fixation Cross Timing: 500 ms Stimulus Pair Timing: 500 ms Probe: Dot probe appeared in the left or right position, remaining until the participant responded Indicate where the dot appeared by pressing one of the two response keys. Probe Timing: Not mentioned Inter-trial interval: 250 ms SOA: Not mentioned Response Timing: Not mentioned Total number of critical trials: 240 Total number of neutral trials: Not mentioned Total number of practice trials: 5	PTSD participants have had greater attentional biases towards the location of cocaine imagery than non-PTSD participants
Bardeen et al. (2013) [25]	In total, 22 cocaine-dependent patients with borderline personality disorder and 36 cocaine-dependent patients without borderline personality disorder	Visual Probe Task	In total, 20 cocaine-related pictures (crack pipes, crack rocks, etc.) and 40 pictures of furniture	Fixation Cross Timing: 500 ms Stimulus Pair Timing: 500 ms Probe: Dot appeared on the screen replacing one of the two pictures. Press a button on the computer keyboard that corresponded to the relative position of the dot on the screen Probe Timing: Not mentioned Inter-trial interval: Not mentioned SOA: Not mentioned Response Timing: Not mentioned Total number of critical trials: 240 Total number of neutral trials: Not mentioned Total number of practice trials: 5	Greater bias for attending to cocaine stimuli among male cocaine-dependent patients with or without borderline personality disorder, when presented with a trauma script intervention
Mayer et al. (2016) [27]	In total, 37 participants randomly assigned to Attention Bias Modification Therapt (ABMT) or control therapy	Visual Probe Task	Cocaine and neutral stimuli were equivalent in size and visual angle, generally matched for colour and content	Fixation Cross Timing: Not mentioned Stimulus Pair Timing: Stimulus pairs presented for 200 ms (speed detection trials) or 500 (difficulty to disengage trials). Probe: Stimulus replaced by a probe (arrow) Probe Timing: Not mentioned Inter-trial interval: 1000 or 1500 ms SOA: Not mentioned Response Timing: 1500 ms Total number of critical trials: 240 Total number of neutral trials: Not mentioned Total number of practice trials: 5	Attention bias modification was not more effective than control at reducing attentional biases

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
