# Peer review of "Variations in the Visual Probe Paradigms for Attention Bias Modification for Substance Use Disorders"

_ijerph, 2019, doi:10.3390/ijerph16183389_

Round 1

Reviewer 1 Report

This is a nice review on possible variations in visual probe clues for possible attention bias modification in the context of substance use illnesses. I do not have any special comments on the manuscritp, it is well written and provides a good overviw on the topic.

It would be even better though if the author could describe some speculative aspects in terms of possible neuroanatomical circuits involved in these variatiosn and how they could be affected by substance abuse disorders. 

Author Response

We thank you Reviewer 1 for peer-reviewing our manuscript and for your positive comments. We have included in our introduction, evidence from previous neuroimaging studies, the neuroanatomical circuits that are involved in attentional biases, and biases modification. The amends are as follows:

“Recent neuroimaging studies have highlighted that attentional biases are associated with increased activation in several neuroanatomical regions, including that of the anterior cingulate cortex, dorsolateral prefrontal cortex, insula, nucleus accumbens and amygdala (26,27).”

Reviewer 2 Report

The manuscript "Variations in Visual Probe Paradigm for Attention Bias Modification for Substance Use Disorders" by Melvyn et al. presents the literature addressing the use of Visual Probe paradigm to alter attention bias in patients with substance (cannibus, opiate, stimulant) use disorders. The authors thoroughly and completely present their findings in Table 1, which would be a valuable resource for researchers as they design their studies. My major concern is the unclear aim of the manuscript.

1) The manuscript should have a central, persistent theme/story. The paper is unfocused-- is the aim to compare methods? is the aim to compare "efficacy" of studies? how is "efficacy" determined (if this is statistical, this needs to be presented in the tables as well)? is the aim to compare internal/external validity? how is that determined? Is the aim to support any of the three theories presented in first paragraph of introduction? Is the aim to compare Stroop with visual probe paradigm? etc. I suggest picking a clear aim, and structuring the paper clearly around this.

2) The paper is difficult to read due to English grammer.

Author Response

We thank you Reviewer 2 for peer-reviewing our paper. We appreciate your recommendations and have made substantial amends to our manuscript.

We have stated in the introduction that there has been, in the published literature, great variation in the stimulus timings used in published studies. We have reframed the aims of our study, to that of comparing the methods and task paradigms of the visual probe task that have been adopted. We hope that this brings better clarity to our paper.

The amends are as follows:

“Given this, our aim is to compare the different task paradigms and methods for the visual probe tasks used for the most highly prevalent substance disorders, that of opiate use, cannabis use and stimulant use disorders.”

We have restructured the paper accordingly and worked on the language of the paper.

Round 2

Reviewer 2 Report

The aim of the study is more clear. There remain punctuation and grammar errors, including but not limited to:

Lines 29-31, 45-46, 49, 67-69, 70, 74, 81...

Also, the term substance use disorder should be consistent throughout the paper (vs. addictive or substance disorder) as per DSM.